# Screening of Cork Oak for Resistance to *Phytophthora cinnamomi* and Micropropagation of Tolerant Seedlings

María Teresa Martínez [1,†], Beatriz Cuenca [2,†], Fátima Mosteiro [1,2], Pablo Piñeiro [1], Felipe Pérez [3], Alejandro Solla [4,*] and Elena Corredoira [1,*]

1    Misión Biológica de Galicia (MBG-CSIC), Sede de Santiago de Compostela, Avda. Vigo s/n, 15705 Santiago de Compostela, Spain; temar@mbg.csic.es (M.T.M.); fmosteir@tragsa.es (F.M.); pablomanuel.pineiro@rai.usc.es (P.P.)

2    Grupo TRAGSA, Viveros Maceda, Carretera Maceda-Balderei, km 2, 32700 Ourense, Spain; bcuenca@tragsa.es

3    Dirección General de Biodiversidad, Bosques y Desertificación, Ministerio para la Transición Ecológica y el Reto Demográfico (MITECO), Avenida Gran Vía de San Francisco 4–6, 28005 Madrid, Spain; fperez@miteco.es

4    Faculty of Forestry, Institute for Dehesa Research (INDEHESA), Universidad de Extremadura, Avenida Virgen del Puerto 2, 10600 Plasencia, Spain

\*    Correspondence: asolla@unex.es (A.S.); elenac@mbg.csic.es (E.C.)

†    These authors contributed equally to this work.

**Abstract:** Massive propagation of cork oak (*Quercus suber*) individuals tolerant to *Phytophthora cinnamomi* (*Pc*) is probably the most important challenge for cork production. Screening for resistance to *Pc* of ca. 200 seedlings obtained from a single cork oak tree that has survived the epidemic was performed by soil infestation. Twenty months after *Pc* inoculation, 33 seedlings survived from *Pc* infection and the four most vigorous seedlings were selected. The plants were forced to produce new shoots under controlled climatic conditions, and the new shoots were used to establish the plants in vitro by axillary budding. High axillary shoot proliferation rates were achieved by culturing the new shoots on Lloyd and McCown (WPM) medium, followed by subculturing for 2 weeks on 0.22 μM benzyladenine (BA) and for 2 weeks further on 0.04 μM BA. Addition of 20 μM silver thiosulphate (STS) increased the proliferation rates and improved the appearance and development of shoots. Rooting rates of 80–100% were obtained by culturing the shoots for 24 or 48 h on Gresshoff and Doy medium with 1/3 macronutrients plus 122.5 μM indole-3-butyric acid and subsequent transfer to root expression medium containing 20 μM STS. The results of this study optimize the micropropagation of a relevant and recalcitrant tree species in forestry.

**Keywords:** axillary shoot proliferation; disease tolerance; genotype; micropropagation; oak decline; *Quercus suber*; rooting; shoot proliferation; silver thiosulphate

## 1. Introduction

Cork oak (*Quercus suber* L.) is a medium sized, slow growing and long-lived evergreen tree native to the Western Mediterranean Basin. Cork oak forests occupy a total area of about 2.2 million hectares in Southwestern Europe (Portugal, Spain, France and Italy) and in northern Africa (Algeria, Morocco and Tunisia) [1]. Cork oak woodlands dominate the southwest of Iberia, where 61% of the world's cork oak forests are found (34 and 27% in Portugal and Spain, respectively) [2]. This species mainly grows in multifunctional (savanna-like) agroforestry systems created and maintained by human management, called *dehesas* in Spain and *montados* in Portugal [3,4]. Due to their high socioeconomic and conservation value, these systems are included in the Special Conservation Areas defined in EU Directive 92/43 [5]. According to European classification criteria, *dehesas* and *montados* are regarded as high natural value farmland due to the importance of the ecosystem services they provide and their role in biodiversity conservation [6].

Cork oak forests are managed for multipurpose uses, although they are mainly exploited for cork production [7,8]. Portugal is the world's largest exporter of processed cork

products (63% of world total), of value equivalent to EUR 1.134 million [9]. Cork oak forests also provide a wide range of other non-wood forest products, including pasture and acorns for livestock, firewood, honey, mushrooms, berries, and aromatic and medicinal plants, and also act as sites for hunting and leisure activities [10]. Forests also provide ecosystem services [11,12] such as carbon sequestration [13], soil and water protection [14] and biodiversity conservation [15]. Finally, cork oak woodlands play a key role in rural development and poverty alleviation in marginal areas where productive options are often scarce.

Nowadays, multiple socioeconomic and biophysical pressures jeopardize the long-term survival of *montados* and *dehesas* [16]. Severe cork oak mortality events have repeatedly occurred in Iberia and Northern Africa, since the 1980s, which has led to significant alteration of the ecosystems [17,18]. Substantial efforts have been made in recent decades to investigate the factors triggering cork oak decline, the interaction of multiple biotic and abiotic factors being the most plausible cause [19]. Extreme weather events in the context of climate change weaken cork oak trees [20] and impact the population dynamics of exotic pathogens [21], making trees more susceptible to diseases [22]. It has been suggested that the oomycete *Phytophthora cinnamomi* Rands (*Pc*) is the main biotic agent triggering cork oak decline [20,23,24]. The effect of *Pc* has been exacerbated by inadequate forest management, including the abandonment of traditional land activities, increased mechanization and stocking rates and the lack of natural regeneration [25,26]. The regenerative capacity of the forests has been weakened by inhibition of recruitment [27], reduced success of reforestation efforts [28] and the recurrence of fires [29].

Genotypes that display tolerance to abiotic and biotic challenges are needed in plantation forestry to counteract losses [30]. A main limitation of forest breeding programs in providing trees that are tolerant to pathogens at present is the lack of knowledge about the transmission of resistance between generations [31]. Identification of tolerant adult genotypes in disease hotspots, selection of tolerant genotypes from their offspring, and vegetative propagation of tolerant plants is necessary for providing the right material in future reforestation programs. In the case of cork oak, the use of acorns from selected parents showing resistance to *Pc* may be a valid approach to increase basal resistance of forests as significant differences in susceptibility to *Pc* have been reported between and within populations and families of *Q. ilex* and *Q. suber* [32,33]. A similar approach has been used to mitigate ash dieback (*Hymenoscyphus fraxineus* (T. Kowalski) Baral, Queloz, Hosoya) [34], pine pitch canker disease (*Fusarium circinatum* Nirenberg and O'Donnell) [35], Dutch elm disease (*Ophiostoma novo-ulmi* Brasier) [36] and *Verticillium* wilt of olive (*Verticillium dahliae* Kleb.) [37]. After identification of 'survivor' adult trees in the field, screening through artificial inoculation of progenies allows selection of further tolerant material and gains in resistance. This methodology has allowed deployment of tolerant trees such as European elms [38], Japanese pines [39] and *Acacia* genotypes [40].

Among the different methods of micropropagation, axillary shoot proliferation from cultured meristems is the most frequently used as it provides genetic stability and is attainable in many plant species, including trees [41,42]. Even in commercial laboratories, micropropagation via axillary shoot proliferation can be used as a routine procedure to propagate several woody species [41,43]. In cork oak, micropropagation by axillary shoot proliferation of axillary buds obtained from juvenile and adult material has been reported [44–47]. However, this method of propagation is rarely used in cork oak nurseries because the protocols published to date show a lack of a stable rate of explant proliferation (i.e., shoot apical necrosis, episodic growth, low proliferation rates or hyperhydric shoots), a genotype-dependent rooting capacity and excessive loss of rooted plantlets during acclimatization due to the low quality of shoots [48].

In view of the interest of landowners and foresters in growing *Pc*-tolerant cork oaks, a national breeding program was started. For the further propagation of the tolerant material obtained, an optimal and rapid protocol for clonal micropropagation is needed. The main objectives of the present study were to (1) screen for *Pc* resistance the progeny obtained from a putatively tolerant cork oak tree, (2) optimize the in vitro shoot establishment and

proliferation steps, (3) assess variation of shoot proliferation within genotypes, and (4) to optimize the rooting step for plantlet regeneration.

## 2. Materials and Methods

### 2.1. Screening for Tolerance to P. cinnamomi

To screen for tolerance, acorns were collected from a single asymptomatic cork oak tree located in an area severely affected by *Pc* in Guadalupe, Extremadura (Spain) (39°25′35.1998″ N 5°17′6.1001″ W). In November 2015, 200 healthy acorns were sampled from the tree, surface disinfected and stratified at 4 °C. In January 2016, the acorns were seeded in root trainers of 300 mL cells filled with peat and vermiculite (Figure 1a), periodically irrigated, and germinating plants were subjected to optimal conditions of growth. In June 2016 and 2017, seedlings were inoculated with *Pc* after inoculum preparation (Figure 1b) as described by Jung et al. [49]. A highly virulent A2 strain of *Pc* (UEx1), isolated from a holm oak in Badajoz, Spain, was used [50]. To increase plant mortality, seedlings were periodically subjected to alternating periods of drought and flooding for two years. Plant mortality was recorded weekly for 20 months after inoculation. In November 2017, to fulfil Koch's postulates, *Pc* was successfully reisolated on PARPH selective medium from the roots of infected seedlings. Final selection of seedlings was based on survival to *Pc* and plant vigor. Vigorous plants in terms of growth usually produce more and longer shoots each season, allowing breeders to propagate them more easily and faster than non-vigorous plants. Moreover, vigorous plants possess competitive ability over conspecifics and other forest species, because of increased access to light, water and nutrients. Small-sized *Q. suber* seedlings with limited root growth for vertical water exploration do not survive summer drought. In December 2017, the surviving seedlings were transplanted into 2 L pots and used for propagation as indicated below. In this study, the selected TGR 123, TGR 128, TGR 144 and TGR 149 cork oak genotypes were used (Figure S1).

### 2.2. In Vitro Establishment Step

To force sprouting, the four tolerant cork oak plants selected were placed in a climatic growth chamber at 25 °C and 80–90% relative humidity under a 16 h photoperiod (90–100 µmol m$^{-2}$ s$^{-1}$ provided by cool-white fluorescent lamps). Three weeks later, new shoots were observed. The sprouting capacity was evaluated by counting the number of new shoots formed per plant and by measuring the length of the shoots. Shoots with their leaves removed were washed with sterile water for 10 min. For surface sterilization, shoots were dipped for 2 min in an aqueous solution of 0.2% sodium hypochlorite (free chlorine) (Millipore, Merck, Darmstadt, Germany), supplemented with 2–3 drops of Tween 80®. Then, the shoots were washed three times with sterile water, for 10 min each time. Sterilized shoots were cut into 0.5–1.0 cm segments. The apexes were discarded and nodal sections only were cultured upright in culture tubes (20 × 160 mm or 30 × 160 mm) containing 20 mL of establishment medium; i.e., Woody Plant Medium (WPM) [51] supplemented with 0.88 µM benzyladenine (BA), 80 mg/L ascorbic acid, 3% sucrose and 0.7% plant propagation agar (PPA; Condalab, Spain).

After the pH was adjusted to 5.7, the medium was autoclaved at 115 °C for 20 min. Ascorbic acid was filter-sterilized before being added to the autoclaved medium. The cultures were placed in a growth chamber with a 16 h photoperiod (50–60 µL m$^{-2}$ s$^{-1}$ provided by cool-white fluorescent lamps at 25 °C) and 8 h of darkness at 20 °C (standard conditions). After culturing for 24 h on establishment medium, each explant was changed to the opposite side of the culture tube to avoid contact with any released phenolic compounds, which may have an inhibitory effect on shoot growth. These initial explants were transferred every 2 weeks to fresh medium of the same composition for 6 weeks. After this period, the following parameters were recorded for each genotype: contamination rate and response rate, defined as the frequency of explants with sprouting buds.

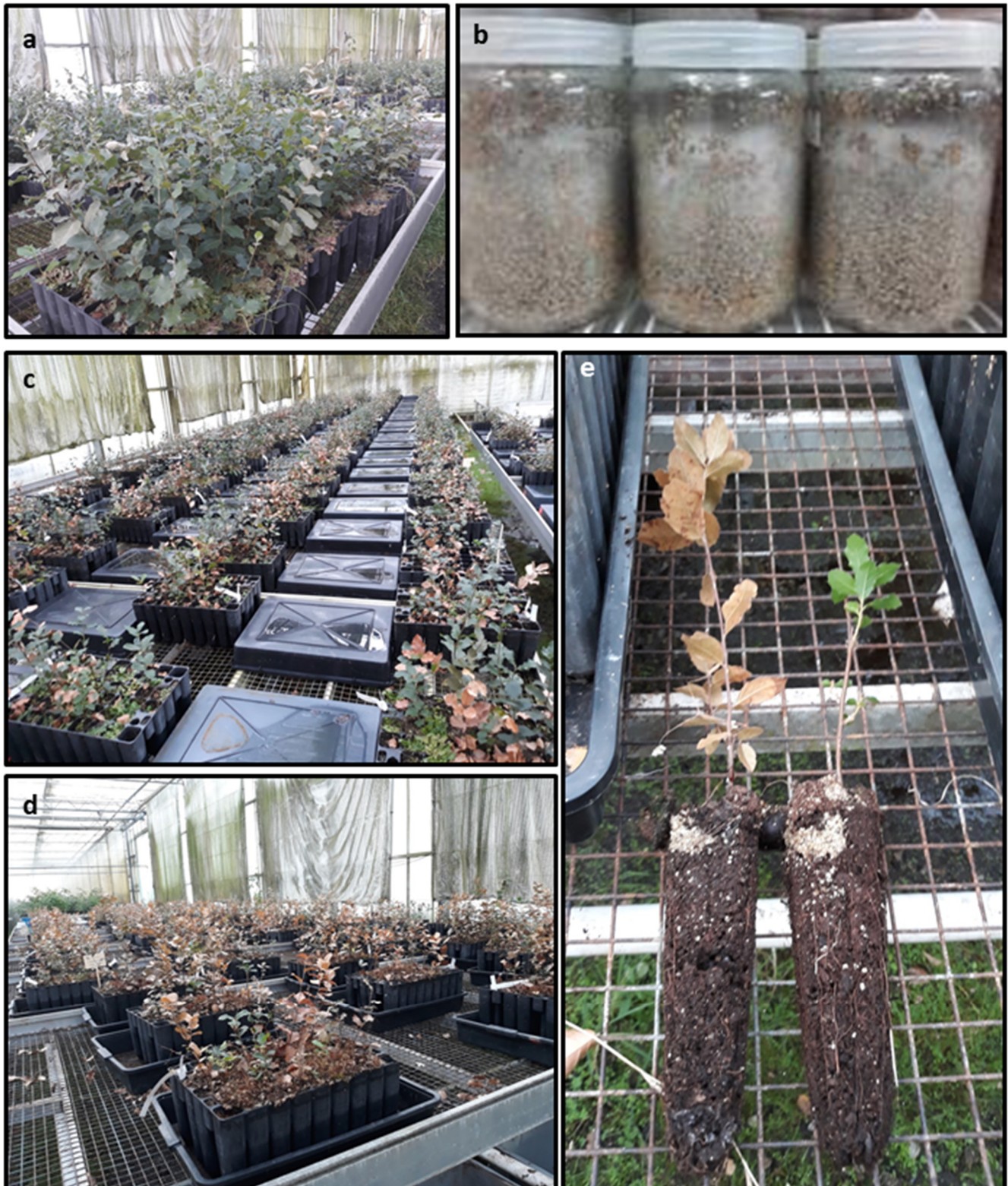

**Figure 1.** Screening for resistance to *Phytophthora cinnamomi* (*Pc*). (**a**) *Quercus suber* seedlings before inoculation. (**b**) One-month-old inoculum of *Pc* grown in 720 mL jars. (**c**) Seedlings showing wilting two months after inoculation. (**d**) Seedlings 20 months after inoculation, at the end of the experiment, with ca. 75% of plants dead. (**e**) *Pc*-susceptible vs. *Pc*-resistant seedlings. Note the inoculum of *Pc* in the top left part of the root ball.

*2.3. Proliferation Step*

After culture of explants for 6 weeks, shoots longer than 5 mm were excised and cultured upright in 500 mL glass jars (90 × 100 mm) containing 70 mL of multiplication medium. This medium consisted of WPM supplemented with 3% sucrose, 0.8% Sigma agar (A-1296; Sigma-Aldrich, St. Louis, MO, USA), 20 µM silver thiosulphate (STS) and a specific cytokinin regime. Shoots were transferred every two weeks over a four-week standard multiplication cycle, by using 0.22 µM BA for the first two weeks and 0.04 µM BA for the last two weeks. STS was filter-sterilized before being added to the autoclaved medium. To achieve the stabilization of shoot cultures, subculturing on multiplication medium with cytokinin regime was performed twice.

To optimize the proliferation step, a first experiment was performed: the multiplication medium used to stabilize shoots was compared with Gresshoff and Doy medium (GD; 1972) [52] to which 3% sucrose and 0.8% Sigma agar were added. As before, shoots were transferred every 2 weeks over a 4-week standard multiplication cycle. GD multiplication medium was chosen on the basis of previous studies of axillary shoot proliferation in cork oak [53]. To assess variation of proliferation between plant individuals, a second experiment was performed: shoots of the four genotypes were cultured on multiplication medium in a 4-week standard multiplication cycle.

In both experiments, at the end of the 4-week multiplication cycle, the following parameters were recorded: proportion of responsive explants (defined as the percentage of explants that formed shoots), number of shoots per responsive explant (considering 0.5–1.0 cm and >1.0 cm shoots), and length of the longest shoot within the responsive explants.

*2.4. Rooting Step*

In the first experiment, apical shoots, 1.0–1.5 cm long, of the four genotypes were cultured in GD medium with macronutrients reduced to one-third strength (¹⁄₃ GD) and 3% sucrose, 0.6% PPA agar and 122.5 µM indol-3-butyric acid (IBA) added. After 24 or 48 h in darkness, shoots were transferred to rooting medium without auxin but supplemented with 20 µM STS. In the second experiment, shoots of the four genotypes were grown on Murashige and Skoog medium (MS; 1962) [54] with half-strength macronutrients (¹⁄₂ MS) and 3% sucrose, 0.6% PPA agar, 14.7 µM IBA and 0.54 µM 1-naphthaleneacetic acid (NAA) added. Shoots were maintained for 14 days in rooting medium: 7 days in darkness and 7 days in light. The shoots were transferred from rooting mineral medium to fresh rooting medium but without auxins but containing 20 µM STS. These rooting treatments were chosen based on previous studies on other oak species [53,55].

The following parameters were determined after the shoots were cultured for 6 weeks: number of shoots that developed adventitious roots, number of roots per shoot, length of the longest root of each shoot, number of shoots with shoot-tip necrosis and number of rooted shoots with secondary root development.

*2.5. Statistical Analysis*

Statistical analysis was performed with Statgraphics Centurion 18 for Windows (version 18.1.14, The Plains, Virginia, USA). Percentage data were subjected to arcsine transformation prior to analysis of variance (non-transformed data are shown in the tables). Means were compared by the least significant difference (LSD) test. In the proliferation experiments, five jars, each containing seven explants per treatment and genotype, were used. In the rooting experiments, 25 shoots were used per genotype and treatment, and each experiment was repeated at least twice.

**3. Results**

*3.1. Screening for P. cinnamomi*

In December 2017, after two inoculations performed in June 2016 and 2017, 33 seedlings survived *Pc* infection (Figure 1c–e). Among the tolerant seedlings, TGR 123, TGR 128, TGR

144 and TGR 149 genotypes were selected as they showed no wilting, were the tallest, and showed straightness and abundance of foliage (Figure S1).

### 3.2. In Vitro Establishment

After 2–4 weeks in the climatic chamber, the four selected plants flushed, although there were some differences regarding the number of new flushed shoots (Table 1). Shoot length fluctuated depending on the genotype, with values ranging from 31.3 to 45.5 cm. The sterilization procedure was efficient and led to a low rate of contamination in initial explants excised from plants, with frequencies ranging from 4.3 to 7.7% (Table 1). After inoculation in the culture medium, the initial explants exuded a brown substance within 24–48 h; however, the addition of ascorbic acid to the culture medium and transfer of the explants to another area of the culture dish appeared to be successful in limiting production of brown exudates (i.e., phenolic compounds released by the explants) and other exudates.

**Table 1.** In vitro establishment of shoot cultures and time to stabilization of shoot proliferation in four *Quercus suber* genotypes tolerant to *Phytophthora cinnamomi*.

| Genotype | Plant Sprouting Ability [a] | | Response to In Vitro Establishment [b] | | | Time to Stabilization of Shoot Proliferation (Months) |
| | Shoots Per Plant | Length of Shoots (mm) | Initial Explants | Contamination Rate (%) | Response Rate (%) [c] | |
| --- | --- | --- | --- | --- | --- | --- |
| TGR 123 | 25 | 31.3 ± 1.0 | 50 | 6.0 | 72.3 | 2.5 |
| TGR 128 | 11 | 41.3 ± 1.1 | 42 | 7.1 | 30.8 | 4 |
| TGR 144 | 23 | 32.2 ± 4.7 | 69 | 4.3 | 48.5 | 4 |
| TGR 149 | 39 | 45.5 ± 17.3 | 104 | 7.7 | 22.9 | 3 |

[a] Evaluated after 4 weeks in a climatic growth chamber. [b] Evaluated after culturing for 8 weeks. [c] Explants with sprouting buds after culturing for 6 weeks in vitro.

The sprouting ability of the seedlings, in terms of shoot number and shoot length, was not correlated with the in vitro bud response in initial explants. Although in vitro response was obtained in all the genotypes tested (Figure 2a,b), there were marked differences in the response frequency, being highest in TGR 123 genotype (72.3%) and lowest in TGR 149 genotype (22.9%) (Table 1).

After culturing the explants for 6 weeks in vitro, new shoots longer than 0.5 cm, formed from initial explants, were isolated and cultured on multiplication medium to check for shoot culture stabilization. After isolation, these shoots proliferated and elongated rapidly during re-culture on fresh medium and showed vigorous growth. Uniform shoot growth was observed within 2.5–4 months, depending on the genotype (Table 1).

### 3.3. In Vitro Proliferation Step

Given the rapid stabilization of the shoot cultures, subculturing on multiplication medium to obtain enough shoots for the proliferation experiments was necessary two or three times only.

In the first experiment, the two mineral media and the addition of STS was assessed on the TGR 123 genotype only (Table 2). WPM medium with STS increased significantly the number of shoots ≥ 1 cm, the total number of shoots and their highest length (Table 2). By using WPM medium, 11.3 shoots 20.6 mm long were obtained, and hyperhydric shoots and apical necrosis were not observed (Figure 2c). In contrast, when using GD medium shoots, hyperhydric shoots and apical necrosis were observed.

In the second experiment, the WPM medium with STS was evaluated in four genotypes. Suitable shoot multiplication was obtained in all genotypes tested, since 100% of explants developed new shoots (Table 3). However, further parameters assessed were genotype-dependent (Table 3). In TGR 123 and TGR 149, the number of shoots ≥ 1 cm was higher than the number of shoots 0.5–1 cm, whereas in TGR 128 and TGR 144 a similar number both types of shoots were obtained. With regard to the total number of shoots, values were significantly higher ($p = 0.0125$) in the TGR 128 genotype. Shoot length ranged from 17.9 to 26.9 mm and the highest values were obtained in TGR 149 genotype (Table 3).

In any clones, using the previous proliferation medium, leaf anomalies (i.e., hyperhydration, folding and chlorosis), shoot-tip necrosis and growth in rosette-like clusters were observed (Figure 2c). The addition of indoleacetic acid (IAA) and NAA to the proliferation medium did not enhance shoot multiplication rates.

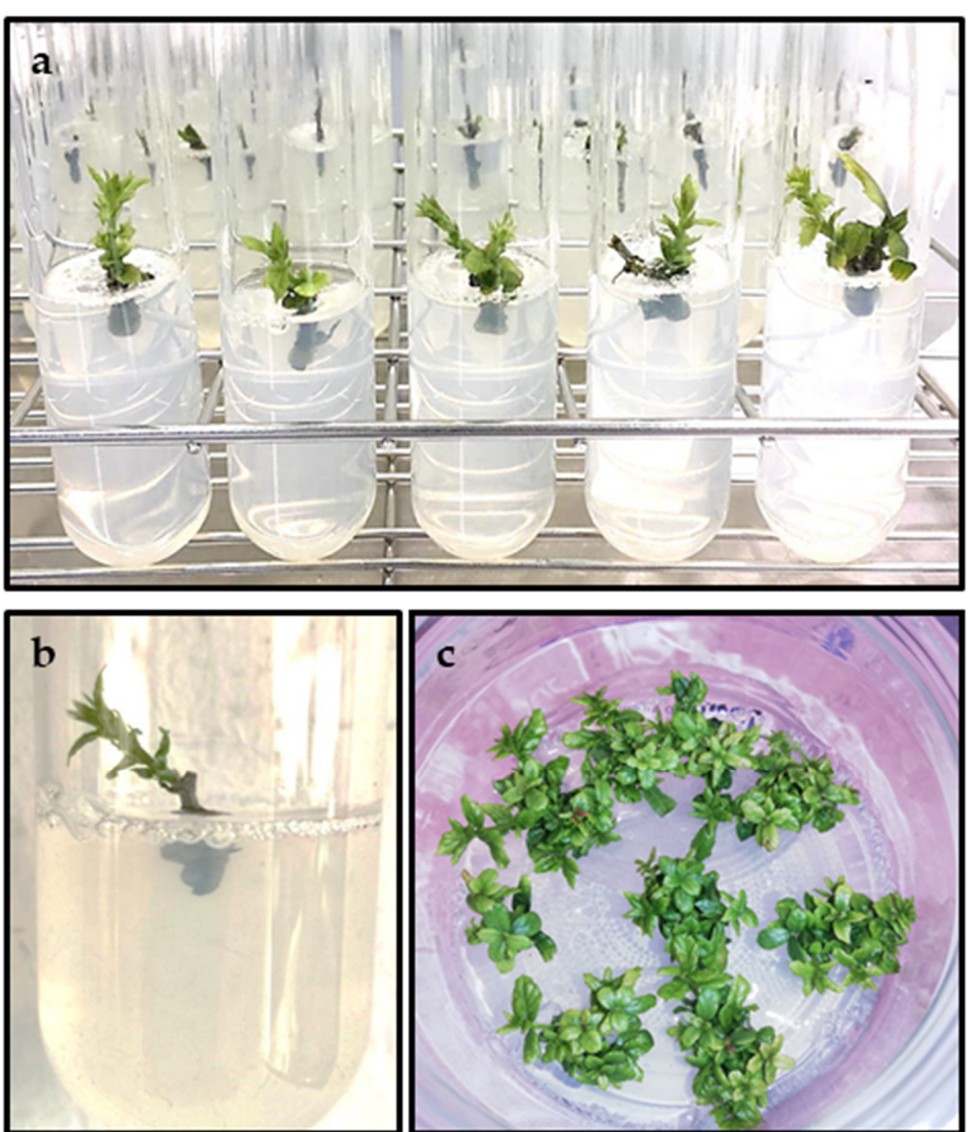

**Figure 2.** Micropropagation of cork oak genotypes selected for their tolerance to *Phytophthora cinnamomi*. (**a,b**) Shoot development from a nodal explant collected from a forced plant. (**c**) Shoots after culturing for 4 weeks on multiplication medium. Tube diameter: 20 mm (**a**); tube diameter: 30 mm (**b**); jar diameter: 90 mm (**c**).

**Table 2.** Effects of two proliferation media and silver thiosulphate (STS) on multiplication of shoot cultures of TGR 123 genotype of *Quercus suber*.

| Medium | Responsive Explants (%) | Shoots 0.5–1.0 cm | Shoots $\geq$ 1 cm | Total Shoots | Longest Shoot Length (mm) |
|---|---|---|---|---|---|
| WPM + STS | 100 ± 0.0 | 3.4 ± 0.2 | 7.9 ± 0.5 b | 11.3 ± 0.4 b | 20.6 ± 0.8 b |
| GD − STS | 100 ± 0.0 | 3.7 ± 0.2 | 3.9 ± 0.5 a | 7.6 ± 0.5 a | 15.0 ± 0.4 a |
| *ANOVA I* | ns | ns | 0.0008 *** | 0.0007 *** | 0.0005 *** |

Values are means ± standard error of five replicate jars, each containing seven explants. ANOVA I significances are shown for each parameter. ns: not significant; *** significant differences at 99.9% ($p \leq 0.001$). Within each column, different letters indicate significant differences at $p$ = 0.05 according to the least significant difference (LSD) test.

**Table 3.** Effect of genotype on proliferation of shoot cultures of four genotypes of *Quercus suber* selected for their tolerance to *Phytophthora cinnamomi*.

| Genotype | Responsive Explants (%) | Shoots 0.5–1.0 cm | Shoots ≥ 1cm | Total Shoots | Longest Shoot Length (mm) |
|---|---|---|---|---|---|
| TGR 123 | 100 ± 0.0 | 3.4 ± 0.3 a | 7.8 ± 0.5 b | 11.2 ± 0.3 a | 20.0 ± 0.6 ab |
| TGR 128 | 100 ± 0.0 | 6.6 ± 0.6 b | 6.0 ± 0.4 a | 12.6 ± 0.4 b | 21.0 ± 1.1 b |
| TGR 144 | 100 ± 0.0 | 6.1 ± 0.3 b | 5.4 ± 0.3 a | 11.5 ± 0.5 ab | 17.9 ± 0.7 a |
| TGR 149 | 100 ± 0.0 | 4.2 ± 0.3 a | 6.1 ± 0.2 a | 10.3 ± 0.4 a | 26.9 ± 0.6 c |
| *ANOVA I* | ns | 0.0003 *** | 0.0028 ** | 0.0125 * | 0.0000 *** |

Values are means ± standard error of five replicate jars, each containing seven explants. ANOVA I significances are shown for each parameter. ns: not significant; * significant differences at 95% ($p \leq 0.05$); ** significant differences at 99% ($p \leq 0.01$); *** significant differences at 99.9% ($p \leq 0.001$). Within each column, different letters indicate significant differences at $p = 0.05$ according to the least significant difference (LSD) test.

### 3.4. In vitro Rooting Step

In all genotypes and treatments, root development was observed (Figure 3). However, rooting rates depended mostly on the treatment (Table 4). In TGR 128, TGR 144 and TGR 149 genotypes significantly higher rooting rates were obtained by culturing in ⅓ GD IBA 122.5 μM than in ½ MS IBA 14.7 μM + NAA 0.54 μM, with no differences between culturing for 24 and 48 h. By contrast, in TGR 123 genotype the highest rates were obtained by culturing in ½ MS IBA 14.7 μM + NAA 0.54 μM, although differences with ⅓ GD IBA 122.5 μM were not significant. Rooting rates were higher than 80% in all genotypes (Table 4). Regarding the number of roots, significant differences were only observed in TGR 149 genotype, with highest number of roots obtained by culturing in ⅓ GD IBA 122.5 μM for 48 h (Table 4). Likewise, in the other genotypes more roots were also obtained with the ⅓ GD IBA 122.5 μM medium. The longest root length was significantly influenced by the treatment, and in all genotypes except in TGR 123, ½ MS 14.7 μM + NAA 0.54 μM rooting medium produced the longest roots (Table 4).

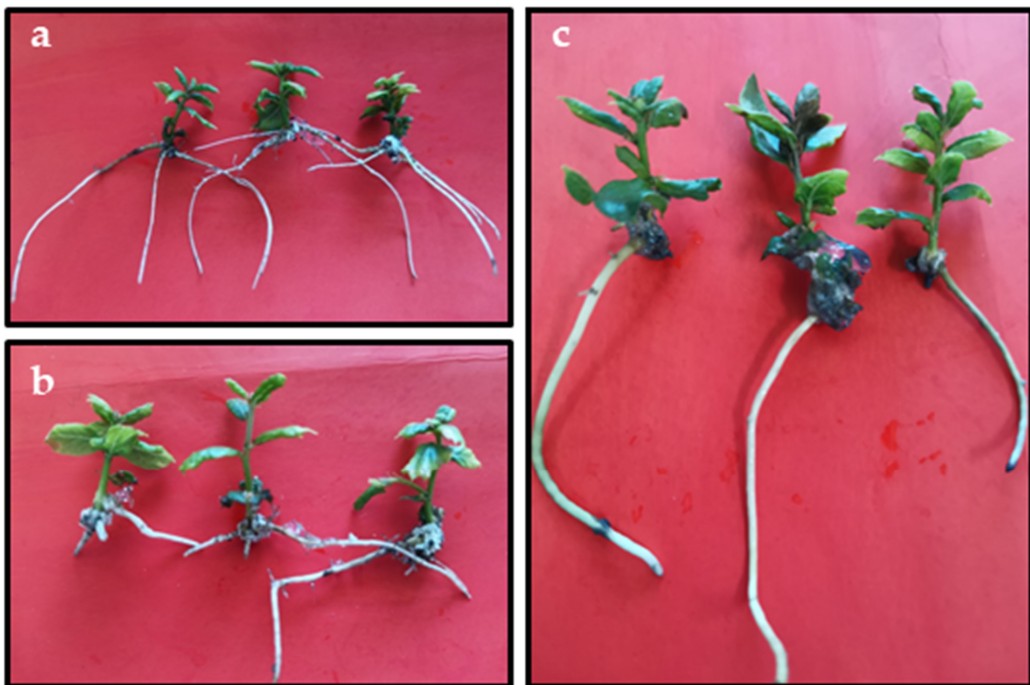

**Figure 3.** Root development on axillary shoots of TGR 123 genotype. (**a,b**) Rooted shoots after culturing 24 h (**a**) and 48 h (**b**) on ⅓ GD rooting medium supplemented with IBA 122.5 μM and 6 weeks further on rooting medium without plant growth regulators. (**c**) Rooted shoots after culturing 15 d on ½ MS rooting medium supplemented with IBA 14.7 μM plus NAA 0.54 μM and 4 weeks further on rooting medium without plant growth regulators.

**Table 4.** Effect of genotype and treatment on the rooting capacity of axillary shoots in *Quercus suber*.

| Genotype | Rooting (%) | Roots (N°) | Longest Root Length (mm) | Shoot-tip Necrosis (%) | Secondary Rooting (%) |
|---|---|---|---|---|---|
| **TGR 123** | | | | | |
| ⅓ GD 24h IBA 122.5 μM | 80.0 ± 7.5 | 3.6 ± 0.3 | 37.7 ± 1.9 | 12.0 ± 5.8 | 73.5 ± 5.1 b |
| ⅓ GD 48h IBA 122.5 μM | 74.0 ± 5.8 | 3.3 ± 0.3 | 40.4 ± 4.6 | 18.0 ± 6.0 | 63.2 ± 9.4 b |
| ½ MS 15d IBA 14.7 Mm + NAA 0.54 μM 122.5 μM mg/L | 84.0 ± 4.8 | 2.6 ± 0.4 | 39.7 ± 5.5 | 34.0 ± 11.4 | 14.5 ± 7.6a |
| *ANOVA I* | ns | ns | ns | ns | 0.0000 *** |
| **TGR 128 \*** | | | | | |
| ⅓ GD 24 h IBA 122.5 μM | 96.0 ± 3.6 b | 2.8 ± 0.3 | 17.2 ± 2.1 a | 0.0 ± 0.0 a | 0.0 ± 0.0 |
| ⅓ GD 48 h IBA 122.5 μM | 96.0 ± 3.6 b | 3.9 ± 0.4 | 16.7 ± 1.3 a | 0.0 ± 0.0 a | 0.0 ± 0.0 |
| ½ MS 15d IBA 14.7 μM + NAA 0.54 μM 122.5 μM mg/L | 76.0 ± 6.7 a | 3.4 ± 0.5 | 40.9 ± 3.2 b | 16.0 ± 3.6 b | 6.7 ± 6.0 |
| *ANOVA I* | 0.0339 * | ns | 0.0000 *** | 0.0004 *** | ns |
| **TGR 144 \*** | | | | | |
| ⅓ GD 24 h IBA 122.5 μM | 100 ± 0.0 b | 3.9 ± 0.3 | 30.4 ± 2.1 a | 0.0 ± 0.0 | 40.0 ± 12.7 |
| ⅓ GD 48 h IBA 122.5 μM | 100 ± 0.0 b | 4.4 ± 0.3 | 24.9 ± 1.1 a | 20.0 ± 9.8 | 5.0 ± 3.5 |
| ½ MS 15d IBA 14.7 μM + NAA 0.54 μM 122.5 μM mg/L | 80.0 ± 8.0 a | 3.8 ± 0.3 | 37.3 ± 1.7 b | 8.0 ± 4.4 | 25.3 ± 10.2 |
| *ANOVA I* | 0.0263 * | ns | 0.0017 ** | ns | ns |
| **TGR 149** | | | | | |
| ⅓ GD 24 h IBA 122.5 μM | 96.0 ± 2.5 b | 3.7 ± 0.4 ab | 23.0 ± 1.7 a | 2.0 ± 1.9 a | 12.0 ± 5.8 |
| ⅓ GD 48 h IBA 122.5 μM | 94.0 ± 2.9 b | 4.2 ± 0.4 b | 21.7 ± 1.6 a | 0.0 ± 0.0 a | 14.0 ± 5.7 |
| ½ MS 15d IBA 14.7 μM + NAA 0.54 μM 122.5 μM mg/L | 62.0 ± 8.7 a | 2.5 ± 0.4 a | 52.1 ± 3.3 b | 18.0 ± 6.6 b | 2.0 ± 1.9 |
| *ANOVA I* | 0.0003 *** | 0.0283 * | 0.0000 *** | 0.0095 ** | ns |

Values are means ± standard error of five explants per jar (five jars were used for TGR 128 and TGR 144 genotypes and ten jars were used for TGR 123 and TGR 149 genotypes). In all genotypes, after rooting treatment the shoots were transferred to rooting medium without auxin, supplemented with STS 20 μM. ANOVA I significance levels are shown for each parameter. ns: not significant; * significant differences at 95% ($p \leq 0.05$); ** significant differences at 99% ($p \leq 0.01$); *** significant differences at 99.9% ($p \leq 0.001$). Within columns, different letters indicate significant differences at $p = 0.05$ according to the least significant difference (LSD) test.

Shoot-tip necrosis rate was relatively low in all genotypes, especially in shoots treated with ⅓ GD IBA 122.5 μM for 24 h. Secondary rooting was low in TGR 128 (0–6.7%) and TGR 149 (2–14%) in contrast to TGR 123 (15–74%) and TGR 144 (5–40%), in which secondary rooting was higher. In all genotypes except in TGR 128, the highest levels of secondary rooting were obtained by culturing on ⅓ GD IBA 122.5 μM for 24 and 48 h (Table 4).

## 4. Discussion

Cork oak populations are severely affected by decline, which has caused mortality of thousands of trees [56]. The use of resistant trees is widely documented to be the safest, most economical and effective procedure for controlling forest diseases [31,35,38,57]. By combining tree breeding with biotechnology techniques, selected trees that are well adapted to changing environmental conditions such as drought, increased temperature, pest and diseases should be quickly and massively generated. This is the case in Spain, where due to the high incidence of cork and holm oak decline, the Ministry of Ecological Transition is supporting a breeding program to identify trees tolerant to drought and *Pc*. Moreover, selected adult cork oak trees are propagated by grafting to create seed orchards, and selected seedlings, after screening for resistance, are cloned to create synthetic varieties.

In the present study, seedlings from one selected cork oak were screened for *Pc* resistance, and quickly micropropagated by axillary budding. In vitro establishment of cultures by axillary budding from the forced sprouting of juvenile plants and adult trees under controlled conditions has been successful in several *Quercus* species [53]. This procedure is highly recommended for in vitro culture in woody species as it allows mild sterilization of the explants, which facilitates their in vitro response. The in vitro response of *Q. suber* explants observed in the present study was better in comparison to the response reported for *Q. ilex* explants [58]. To date, the main limitations of in vitro establishment step in cork oak have been the browning of explants and finding the right medium [46,48].

Changing the position of explants in the culture tube and periodic transfer to fresh medium during the establishment period are effective strategies for preventing the negative effect of the release of phenolic compounds by explants after sterilization. This procedure is relevant and has been scarcely mentioned in the literature.

The selection of an appropriate culture medium is of paramount importance to promote healthy growth of shoots [59]. The mineral composition of media can vary depending on the culture step involved; i.e., establishment, multiplication and rooting [42,60]. The proliferation medium used in the present study was based on a previously defined medium used for multiplication of axillary shoots of juvenile and adult genotypes of holm oak [55,58]. As in holm oak, WPM medium and STS improved the proliferation rates and shoot quality of cork oak, in terms of shoot length, if compared to GD medium without STS. Mineral salts are known to affect axillary shoot proliferation capacity, which varies between species and cultivars [42]. In cork oak, WPM medium has seldom been used for proliferation of axillary shoots. A previous study [45] concluded that mineral media with low concentrations of ions, such as Sommer's medium [61] or Heller [62], are most suitable for the growth and proliferation of cork oak shoots. Subsequently, in [46] the GD medium in adult trees was used. In contrast, [63] used MS medium for axillary shoot proliferation of cultures derived from 4- to 9-month-old *Q. suber* plants. Later on, [47] proposed combining WPM macronutrients and MS micronutrients in the mineral medium to obtain adequate shoot proliferation in shoots derived from embryonic axes. However, in other oak species as, for example, in *Q. rubra* [64], WPM was the best medium for proliferation of shoots cultures. During the vitro establishment of 12 oak species, better growth responses and longer survival times were observed in explants grown on WPM than in explants grown on GD medium [65]. Finally, in *Quercus lusitanica,* higher multiplication rates were observed with WPM than with GD medium [66].

The major difficulties regarding achieving adequate multiplication rates include the presence of phenolic compounds that can be released into the culture medium and that reduce or even inhibit proliferation, shoot-tip necrosis and hyperhydricity in the shoots [48]. Hyperhydricity (also formerly known as vitrification) is a morphological and physiological disorder of plants vegetatively propagated in vitro including a set of physical characteristics in leaves, stems and roots that give them a crystalline, moist or watery, translucent appearance [67,68]. This disorder affects photosynthesis and gas exchange and makes the shoot unsuitable for rooting. The presence of vitrified shoots in cultures greatly decreases the productivity. It should be noted that in the present study, the incorporation of 20 $\mu$M STS in the proliferation medium prevented hyperhydricity and shoot-tip necrosis. During in vitro culture, dividing cells produce ethylene, which acts as a growth inhibitor causing many of the aforementioned disorders. Chemical compounds containing silver ions are able to block the action of ethylene [69] and consequently promote the growth of plant cultures and reduce leaf abscission, hyperhydricity and shoot-tip necrosis [70]. Among the different compounds used to reduce the effect of ethylene, STS has become popular in recent years because of its mobility and low phytotoxicity [71]. For example, in *Centella asiatica*, the addition of STS had a positive effect on shoot regeneration percentage and number of shoots [72]. Similar results were also obtained in [55], which improved the proliferation capacity and morphological appearance of axillary shoots derived from adult holm oak trees by including 20 $\mu$M STS in the medium. In cork oak, the addition of 20 $\mu$M STS to the WPM medium resulted in efficient proliferation and controlled hyperhydricity and shoot-tip necrosis.

The most widely used cytokinin in axillary shoot proliferation of cork oak is benzyladenine, but the addition of low concentrations of the auxin naphthaleneacetic acid improves multiplication rates, especially in material of adult origin [48]. By contrast, in the present study, the inclusion of IAA or NAA did not enhance the proliferation rates. Other cytokinins such as zeatin have been used in combination with BA for shoot proliferation of oak species [53]. In cork oak, the use of zeatin was discarded as it is expensive and good results in terms of number and quality of shoots were obtained with BA alone.

Shoot proliferation values obtained were higher than those mentioned in the literature for other oak species such as *Q. alba*, *Q. bicolor* and *Q. rubra* [73]. In our four tolerant individuals, the shoot proliferation rates were high and varied depending on the genotype. The effect of genotype on in vitro behavior is well documented in the literature on oak and other forest species. For example, [64] reported variation in multiplication rates in nine clones of *Q. rubra* (six of juvenile origin and three of adult trees) owing to the genotype and age. In the same way, [58] reported that proliferation rates of shoot cultures derived from *Pc*-tolerant plants of *Q. ilex*, using the same medium as here, were genotype dependent.

The rooting rates of microcuttings are influenced by different factors such as the origin of cuttings (juvenile or mature), exudation of inhibitory compounds, genotype, accumulation of ethylene during in vitro rooting, and the type of auxins and their concentration and combination [53,74,75]. In the present study, despite the recalcitrance of cork oak, by applying high concentration of IBA for a short period, rooting rates higher than 80% were obtained. Exposure of cork oak shoots to low concentrations of IBA or dipping their basal ends in a concentrated solution of IBA for 1–2 min were methods that previously provided the best results [45,47]. One limiting factor during the rooting step was the appearance of shoot-tip necrosis as a consequence of auxin treatment. This problem frequently occurs during oak micropropagation, with the terminal buds of newly developed shoot dying in culture [53]. As in the proliferation step, in the present study the incorporation of STS to rooting expression medium improved rooting rates and reduced shoot-tip necrosis in cork oak shoots. Likewise, in *Corymbia maculate,* root induction and growth of shoot tips cultured were enhanced by inclusion of STS in the culture medium [76]. In a previous study [77], the addition of 10 µM STS to the culture medium stimulated the development of buds and roots in shoot cultures of *Colobanthus quitensis* and also inhibited yellowing and death of leaves. Moreover, in holm oak, the rooting rates of shoots derived from juvenile plants was improved by adding 20 µM STS to the rooting expression medium [58].

The long-term stability of *Pc* resistance in the propagated cork oak individuals is ignored. According to our observations, during the juvenile stage the tolerance of cork oak to *Pc* increases with age (unpublished results). To check for the adaptation and stability of *Pc* resistance of the selected material, experimental field plots similar to those used in elm [38] will be established. These field plots will be located in contrasting locations in Spain [78] to allow exploring phenotypic plasticity of *Pc* resistance, and durable resistance across different environmental conditions, *Pc* strains and tree ontogenetic stages [79].

## 5. Conclusions

The culture and plantation of tolerant trees is considered the most efficient and environmentally friendly strategy for controlling forest diseases [79]. A first step for breeding trees for resistance is to identify tolerance in mother trees. As cork oak is a recalcitrant species, the tolerance of a 'survivor' cork oak tree (present in a *Pc* center) was checked here through its progeny. Moreover, by germinating acorns and by inoculating plants with a virulent *Pc* strain, four resistant genotypes were selected. The procedure used here could serve as protocol to be applied in other species. Obviously, more mother trees putatively tolerant to *Pc* should be screened in order to increase the genetic basis of the selected seedlings in terms of resistance and adaptability to unfavorable environments. Conservation and propagation of the selected tolerant seedlings was carried out by axillary budding after refinement of the sterilization step and optimization of the proliferation and rooting steps. The most suitable mineral medium and the best timing and auxin concentration were revealed. The use of STS, added here to the culture medium for the first time in cork oak, improved significantly the proliferation and rooting rates and also reduced hyperhydricity and shoot-tip necrosis. The results obtained provide relevant and new information on cork oak micropropagation by axillary budding.

**Supplementary Materials:** The following supporting information can be downloaded at: https://www.mdpi.com/article/10.3390/horticulturae9060692/s1, Figure S1: Appearance of cork oak tolerant plants selected by *Pc* screening before culture in the climatic chamber.

**Author Contributions:** Conceptualization, B.C., A.S., E.C., F.P. and M.T.M.; investigation, B.C., A.S., F.M., E.C., P.P. and M.T.M.; writing—original draft preparation, M.T.M., E.C. and B.C.; writing—review and editing, M.T.M., B.C., E.C., A.S. and F.P.; supervision, E.C. and M.T.M. All authors have read and agreed to the published version of the manuscript.

**Funding:** This research was funded by the Spanish Ministry for the Ecological Transition and the Demographic Challenge (MITECO) and 75% co-funded by the National Programme for Rural Development 2014–2020 through the European Agricultural Fund for Rural Development (EAFRD), sub-measure 15.2 "Support for the conservation and promotion of forest genetic resources". Previous screening was funded by the Spanish Ministry of Economy and Competitiveness (RTA2014-00063-C04-04).

**Data Availability Statement:** Not applicable.

**Conflicts of Interest:** The authors declare no conflict of interest.

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
