# Peer review of "Screening of Cork Oak for Resistance to Phytophthora cinnamomi and Micropropagation of Tolerant Seedlings"

_horticulturae, doi:10.3390/horticulturae9060692_

Round 1

Reviewer 1 Report

This study provides optimized micropropagation techniques for cork oak, a valuable and challenging tree species in forestry, allowing for the mass propagation of Pc-resistant individuals, which could contribute to the sustainable production of cork. The paper is well written and can be accepted after the minor adjustment.

1. Keywords are not appropriately selected; Line 53; add the year and also provide the web link; Also add the categories of this species as per IUCN guidelines.

2. Long-term resistance stability - The author also need to discuss on the long-term stability of Pc resistance in the propagated cork oak individuals. It is essential to investigate whether the resistance traits remain stable over successive generations and under different environmental conditions.

3. The protocol relies on screening seedlings obtained from a single cork oak tree that has survived the Pc epidemic. This narrow genetic base may result in limited diversity among the propagated individuals, which could impact their long-term adaptability and resilience. Author need to discuss these issue as well.

4. Potential for reduced vigor: Although the protocol selects the four most vigorous seedlings, there is a possibility that the chosen individuals may not possess the full range of desired traits, including growth rate, resilience to other stressors, or overall fitness. This could limit the overall performance of the propagated cork oak individuals. How the author will encounter this?

5. What is the purpose of putting in vitro shoot on rooting medium supplemented with the auxin followed by 6 wk on hormone free medium?

6. What factors contribute to the varying levels of shoot tip necrosis and secondary rooting among different genotypes of cork oak when treated with different concentrations of GD IBA (indole-3-butyric acid) and durations of exposure?

7. Also add the study related to acclimatization and establishment of the plant.

NA

Reviewer 2 Report

Dear Authors, congratulations on your manuscript which is well organized and scientifically appropriate. Contributing to genetic improvement through field selection methodologies and subsequent innovative multiplication is the most sustainable pathway both environmentally and economically with not insignificant social implications. In vitro propagation was well articulated and structured on the most innovative methodologies.

At this stage I would like to ask a question and make a suggestion:

1. has there been an assessment of the presence of PC in acclimated seedlings after in vitro propagation? That is, has any occurrence of mutagenesis induced by in vitro propagation been verified?

2. the 'conclusions' section has been formulated as if it is an abstract and in my opinion requires a different approach and content with more relevance to possible future in-depth research, especially regarding the in vivo growth activity of these selected and improved genotypes and the stability of the character of resistance or tolerance to PC.

Reviewer 3 Report

Dear editor/author

The present manuscript presents clearly, very useful data for the exploitation/consrevation of Cork oak forests that are managed for multipurpose uses.

The selection of tolerant seedlings was applied and a suitable micropropagation protocol is presented for tolerant clones.

The methods are clearly described, and the statistical evaluation is OK. 

The data is clearly presented and there are representative photos of all the experimentation stages.

The conclusions are clearly defended by the data in the results and the discussion section.

However, I think that the authors could discuss more the difference between suitable clones. Are all the clones of high multiplication rate? Does the research release the impact of one or two suitable clones? I think that it is very important for the present manuscript. Furthermore, the first paragraph of the discussion is relevant to the background of the research, and I think it could be moved in the Introduction. Some more research on the effectiveness of zeatin could be very useful for the present research. 

Regarding the literature, there are recent references, the reference list being extensive.

The article adds significant data to the current literature, and it could be presented as a printed manuscript in the present SI of Horticulturae after major revision.  

In the attached .pdf there are some more issues.

Author Response

The authors thank, with gratitude, the effort carried out by the Reviewer in order to correct and improve the quality of the manuscript. Here are the responses to the Reviewer queries.

The present manuscript presents clearly, very useful data for the exploitation/conservation of Cork oak forests that are managed for multipurpose uses. The selection of tolerant seedlings was applied and a suitable micropropagation protocol is presented for tolerant clones. The methods are clearly described, and the statistical evaluation is OK. The data is clearly presented and there are representative photos of all the experimentation stages. The conclusions are clearly defended by the data in the results and the discussion section.

1. However, I think that the authors could discuss more the difference between suitable clones. Are all the clones of high multiplication rate? Does the research release the impact of one or two suitable clones? I think that it is very important for the present manuscript.

Although there are differences between the clones in the proliferation ability, in general the values obtained can be considered high taking into account that cork oak is a recalcitrant woody species. This part of discussion has been improved in the new version (see lines 426-430).

2. Furthermore, the first paragraph of the discussion is relevant to the background of the research, and I think it could be moved in the Introduction.

Thank you for your suggestion, but in our opinion that paragraph is more suitable for discussion.

3. Some more research on the effectiveness of zeatin could be very useful for the present research.

This information was added in the new version (see lines 423-425).

Regarding the literature, there are recent references, the reference list being extensive.

The article adds significant data to the current literature, and it could be presented as a printed manuscript in the present SI of Horticulturae after major revision.

Please see the attachment for other comments.

Reviewer 4 Report

The manuscript submitted by Martínez et al. describes the screening and propagation of Pc resistant genotype cork oak tree. The topics described are interesting and valuable for wide range of readers. The results are clearly presented and the conclusions are hardly controversial.

Minor comments

In some of the Discussion sections, the citation of references seems to be incorrect, e.g., lines 341 and 342. They need to be corrected.

Round 2

Reviewer 3 Report

Dear Editor/Author

Authors responded to my suggestions.

I believe that the present article adds significant data to the current literature, and it could be presented as a printed manuscript in the present SI of Horticulturae.

Kind regards